# iAcety–SmRF: Identification of Acetylation Protein by Using Statistical Moments and Random Forest

**DOI:** 10.3390/membranes12030265

**Published:** 2022-02-25

**Authors:** Sharaf Malebary, Shaista Rahman, Omar Barukab, Rehab Ash’ari, Sher Afzal Khan

**Affiliations:** 1Faculty of Computing and Information Technology, King Abdulaziz University, Jeddah 21911, Saudi Arabia; smalebary@kau.edu.sa (S.M.); obarukab@kau.edu.sa (O.B.); rashary@kau.edu.sa (R.A.); 2Department of Computer Science, Abdul Wali Khan University Mardan, Mardan 23200, Pakistan; sher.afzal@awkum.edu.pk

**Keywords:** acetylation, random forest, probabilistic neural network, statistical movement, post-translational modification, machine learning, membrane proteins

## Abstract

Acetylation is the most important post-translation modification (PTM) in eukaryotes; it has manifold effects on the level of protein that transform an acetyl group from an acetyl coenzyme to a specific site on a polypeptide chain. Acetylation sites play many important roles, including regulating membrane protein functions and strongly affecting the membrane interaction of proteins and membrane remodeling. Because of these properties, its correct identification is essential to understand its mechanism in biological systems. As such, some traditional methods, such as mass spectrometry and site-directed mutagenesis, are used, but they are tedious and time-consuming. To overcome such limitations, many computer models are being developed to correctly identify their sequences from non-acetyl sequences, but they have poor efficiency in terms of accuracy, sensitivity, and specificity. This work proposes an efficient and accurate computational model for predicting Acetylation using machine learning approaches. The proposed model achieved an accuracy of 100 percent with the 10-fold cross-validation test based on the Random Forest classifier, along with a feature extraction approach using statistical moments. The model is also validated by the jackknife, self-consistency, and independent test, which achieved an accuracy of 100, 100, and 97, respectively, results far better as compared to the already existing models available in the literature.

## 1. Introduction

Proteins are the basic and key part of the human body that perform many kinds of major functions in and outside a cell. The proteins are translated or synthesized from messenger RNA which is first codified into ribosomes and makes a chain of amino acid or polypeptide. After the translation process, certain amino acids can experience chemical changes at the protein’s C-termini or N-termini or in amino acid side chains, known as post-translation modifications (PTLM or PTM). The PTM can modify or may introduce the new functional group to the protein, such as in Acetylation, an example of Acetyl-lysine is shown in the Figure 1) [1]. It plays a key role in making protein products [2,3,4]. Each protein in the proteome may be altered either before or after it is translated. The charge state, hydrophobicity, conformation, and stability of a protein are all affected by various changes, which, in turn, influence its function. Protein modification has a variety of functions in different organs: (1) It ensure the fast and complex response of cells to regulate intra-cellular communication, division, and growth of cells (2) also pivotal for various physiological and pathological mechanism.

Protein acetylation can be achieved using a variety of methods, this adds the acetyl functional group into a chemical compound, which make another ester, the acetate.

Another form of lysine residue is usually acetylated [1,2,3,4]. The active substance, acetic anhydride, is commonly used to react with free hydroxyl groups as an acetylating agent. It is used in, for example, aspirin, heroin, and THC-O-acetate synthesis. Thousands of acetylated mammalian proteins have been identified [1], in addition to protein analysis. The acetylation takes place, for example, via a co-translation and a post-translational adaptation of proteins, histone, and p53, in addition to tubulins. Among these proteins, there is a high representation of chromatin proteins and metabolic enzymes, suggesting that acetylation, in addition to digestion, has an extremely important influence on the appearance of the genetic material. Among the microbes, 90 percent of the proteins that are surrounded by the central metabolism of Salmonella Centrica are acetylated [1,2].

Further, acetylation sites also play an important role in regulating membrane protein functions of multiple families, as documented in Reference [5]. It is supported by examples that acetylation is significantly enhanced in membrane-binding regions, where it is often located directly in critical membrane-binding pockets ideally positioned to modulate membrane interactions. Moreover, it was found that acetylation and acetylation mimetics strongly affected the membrane interaction of proteins, resulting in decreased membrane affinity and, in the case of amphiphysin and EHD2, altered membrane remodeling [6]. In cells, mimicking even a single acetylation event within the membrane interaction region reduced the binding affinity to membranes, resulting in cytoplasmic dispersal. In another report, acetylation affected the effects on membrane interaction, as well as membrane remodeling [7]. Similarly, the ability to control the membrane-binding activity of C2 domains via acetylation could allow the cell to further regulate Ca-dependent transmembrane transport and signaling events. It has also been documented that acetylation is present on the membrane-binding surface of the phosphatase domain of K163/K164, as it appears to be that Alanine mutations reduce membrane binding [8]. In addition, two reports on proteins with PH domains indicate that acetylation has opposite effects on membrane localization in cells (either increased or decreased) [9,10]. Much has been learned about the acetyltransferases and deacetylases that regulate protein-DNA-protein-protein interactions [2,5,11]. Some of these enzymes may also be involved in controlling protein membrane interactions. Consistent with this idea, localization of acetyltransferase in the cytoplasm and deacetylases in cell membranes have been observed [12,13,14].

Acetylation also plays a prominent role in numerous important cellular processes, such as stability and localization of protein [4,5]. In addition, modification of S/T/Y sites by acetylation, glycosylation, sulfation, and nitration has been reported [5,6]. Moreover, it also plays a role in the modulation of gene expression by histone alteration, as well as is a very significant function in controlling cellular metabolism and protein folding [15,16,17,18].

From the above discussion it revealed that Acetylation is an important post-translation modification, and it is necessary to correctly identify them; however, it remains a major challenge to understand the functions and regulations of the molecular acetylation mechanism. Many traditional approaches are in use for their identification, including high-throughput mass spectrometry (MS) [19,20]. However, since the acetylation mechanism is complicated, rapid, and reversible, such methods remain time-consuming, expensive, and laborious [21,22].

To overcome the existing problems in its identification, many researchers have developed a computational model for fast and inexpensive prediction of PTM sites [15,16,23,24], such as the ubiquitination [17,18,24], the phosphorylation [15,16,19,25,26,27], sumoylation [28,29,30], and the acetylation [31,32,33,34], etc. The important step for the PTM prediction model to correctly transform the biological sequences into their equivalent numerical form, for this purpose, many feature extraction methods are developed which are documented like the amino acids composition (AAC), the dipeptides composition (DPC), and Pseudo Amino Acid Composition (PseAAC) [35]. For such feature extractions many methods are discussed in Reference [20].

Reference [1] proposed a novel measurement procedure iAcet-PseFDA, a classification model for acetylation proteins by extracting features come from sequence conservation information using a gray structure model and KNN scoring based on functional domain annotation databases including GO [36] and subcellular localization for acetylation protein recognition. The authors achieved 77.10 percent accuracy using 5-fold cross-validation on three datasets, with a significant amount of attribute analysis and the discovery algorithm for relief functionality.

Reference [37] proposed a method ProAcePred to predict prokaryote-specific lysine acetylation sites, using SVM, 10-fold cross-validation, and the elastic net mathematical approach for optimizing the dimensionality of feature vectors, which greatly increased prediction accuracy and yielded promising results.

Wuyun et al. [38], developed a model, KA-predictor, to predict species-specific lysine acetylation sites using the classifier SVM. They achieved highly competitive for the majority of species as compared with other methods.

Hou [39] suggested a predictor for lysine acetylation prediction called LAceP, based on logistic regression classifiers and various biological characteristics. Using Random Forest classifiers, Li [40] developed SSPKA, a tool for species-specific lysine acetylation prediction.

From the above discussion, it has been observed that many predictors have been developed for the identification of acetylation sites; however, the maximum prediction accuracy established in all previous models was 77.10%, which is very poor for a correct identification of the acetylation sites.

Therefore, in the present study, we use statistical moments as feature extractions and Random Forest and PNN as a classifier. Further, the model evaluation is done by 10-fold cross-validation, self-consistency, independent, and by jackknife testing. We obtained dominant results as compared to the existing models that were developed earlier.

Therefore, to improve the predictor model, we use statistical moments as feature extractions and Random Forest and PNN as classifiers. The model evaluation is carried out through 10-fold cross-validation, self-consistency, independently, and the jackknife tests. We achieved dominant results as compared to the existing models developed earlier.

## 2. Materials and Methods

In this review, we follow the 5-step procedure mentioned in References [16,41] to establish a predictive predictor for biological sequences. It consists of the following steps: (1) choosing or generating an appropriate benchmark dataset to be used for training and testing; (2) transforming a biological sequence into its equivalent mathematical form, which returns the basic correlation with the biological sequence, and transforming a biological sequence into its equivalent mathematical form, which returns the required correlation with the biological sequence; (3) developing or using existing classifier for the required predictor; (4) using cross-validation experiments to determine the accuracy of the suggested indicator; and, finally, (5) developing an influential website/GitHub resource for public use for the benefit of future research and development. All the above steps are presented in Figure 2. Further, details of the above can be found in the following subsections.

## 3. Benchmark Dataset

We begin with collection of a valid benchmark dataset for training and testing, which is the first step in the 5-step rule [1]. The dataset is collected from the well-known data repository, the UNIPROT http://www.uniprot.org, retrieved dated 8 December 2021. It contains 2900 protein samples, of which 725 were positive denoted by S_posi_, and 2175 were negative denoted by S_negt_. Further, for the effectiveness of the proposed predictor, the set of negative data samples is equally divided in three sets, ${\bar{S}}_{1}$, ${\bar{S}}_{2}$, and ${\bar{S}}_{3}$ as in Reference [1] such that,
$${\bar{S}}_{i}\cap {\bar{S}}_{j}=\phi ,\;\;\;\;\;\;\left(i\neq j;,\;i,\;j=1,\;2,\;3\right),$$
where the sign ∩ represents the intersection of sets. Furthermore, we individually combine these three negative datasets ${\bar{S}}_{1}$, ${\bar{S}}_{2}$ and ${\bar{S}}_{3}$ with S_posi and created three new datasets with the same number of positive and negative samples as expressed in Equation (1) below:(1)$$(S_{\mathrm{posi}}\cup {\bar{S}}_{1}),\;\left(S_{\mathrm{posi}}\cup {\bar{S}}_{2}\right),\;(S_{\mathrm{posi}}\cup {\bar{S}}_{3}).$$

The symbol ∪ denotes the union of two sets. It is important to note that the dataset under discussion was used for prediction by [1], whereas the positive dataset was collected based on the given three steps: (1) The possible proteins to be acetylated are identified by a single fixed keyword, i.e., {N acetylcysteine, N acetylserine, N acetylglutamate, N acetylglycine, N acetylcholine, N acetylthreonine, N acetylcholine, N acetylmethionine, N acetylmethionylc, N acetyltyrosine, O-acetylserine, or N6-acetyllysine O-acetyltheronine O}; (2) here, protein collection was validated using some assertion technique; and (3) the 30, or additional, amino acids in proteins, along with the redundant proteins, were removed as discussed in [1].

Whereas the negative data was generated in a similar way to the positive data, except those proteins are not a member being searched by the above keywords? As a result, it produced a large number of negative samples, moreover, random selection is made from those that were balanced in size to positive samples.

## 4. Feature Extraction

The existing traditional classifiers, such as SVM, KNN, ANN, and many others, are not as powerful in classifying the biological data and making the required prediction. Therefore, a medium is needed to convert biological data into the necessary numerical form to make it suitable for traditional classifiers. For this purpose, many models are developed to extract the required characteristics from biological data, e.g., PseAAC, AAC, Pse-in-One, Pse-Analysis, and many more [27,28,29]. In feature extraction, the emphasis is on preserving the critical properties of the protein, its location, and functions. The statistical moment [42] is used to derive features in this study, which is discussed in detail below.

### 4.1. Statistical Moments

In statistics and probability distributions, some form remains beneficial when performing analysis of a particular sequence. The study of such configuration of data collection in pattern matching is known as moments [25]. There are useful moments when there are various pattern recognition problems related to feature development that do not depend on the pattern or sequence parameters provided [27,29,30,31,43]. Particular moments are used to calculate data size, data alignment, and data eccentricity. In this study, we extract the necessary features of acetylation proteins using Hahn, raw, and central moments. The raw moment is used to estimate the probability distribution by using mean, variance, and asymmetry, these moments are neither location invariant nor scale invariant [32]. Similarly, the same procedure is used in case of the central moment, but the calculation is based on the data centroid. This moment is a scale variant and location invariant. The Hahn moments, on the other hand, are dependent on Hahn polynomials, it is neither scale invariant nor location variant [33,34,44]. These moments are very important for extracting obscure features from protein sequences, as they contain complex orderly details about biotic sequences. In the proposed work, a linearly planned structural of a protein sequence is used, as given in Equation (2).
(2)$$P=R_{1}R_{2}R_{3}\ldots R_{L},$$
where $R_{1}$ is the 1st amino acid, represented in proteins P, the last amino acid is $R_{L}$, and total length is ‘L’. Transforming the information of the protein linear structure as given in Equation (2) into 2D matrix representation of dimension k as computed by the following equation.
k = [√P](3)
where “P” represents the protein sequence length, and k represents the dimension of the obtained 2D square matrix.

Hence the Equation (4), represents the matrix denoted by $N^{\prime }$ is constructed by using the order obtained from Equation (3) that is k × k
(4)$$N^{\prime }=[\begin{array}{cccccc} T_{1\to 1} & T_{1\to 2} & \cdots & T_{1\to j} & \cdots & T_{1\to k}\\ T_{2\to 1} & T_{2\to 2} & \cdots & T_{2\to j} & \cdots & T_{2\to k}\\ \vdots & \vdots & \cdots & \vdots & \cdots & \vdots \\ T_{i\to 1} & T_{i\to 2} & \cdots & T_{i\to j} & \cdots & T_{i\to k}\\ \vdots & \vdots & & \vdots & & \vdots \\ T_{k\to 1} & T_{k\to 2} & \cdots & T_{k\to j} & \cdots & T_{k\to k} \end{array}]$$

The raw moment R (a, b) is computed by the values of N’, which is a continuous 2D function of order (a + b), as shown in Equation (5):(5)$$R_{\mathrm{ab}}=\sum_{p}\cdot \sum_{q}\cdot p^{a}q^{b}N^{\prime }\left(p,q\right).$$

Up to order 3, the equation calculates the raw moments. This raw moments are measured using the data’s roots as a starting point [45,46,47,48]. R_00_, R_01_, R_10_, R_11_, R_02_, R_20_, R_21_, R_30_, and R_03_ are the raw moment’s characteristics, weighed up to order 3rd.

The centroid is a point from where all points are equivalently dispersed in all directions with a weighted average [45,48,49,50]. The following equation, which uses the centroid, calculates the special characteristics of central moments up to order 3 (6).
(6)$$C_{\mathrm{ab}}=\sum_{p}\cdot \sum_{q}\cdot {\left(p-\bar{p}\right)}^{a}{\left(q-\bar{q}\right)}^{b}N^{\prime }\left(p,q\right).$$

The unique features are calculated up to 3rd order as: C_00_, C_01_, C_10_, C_11_, C_02_, C_20_, C_30_, and C_31_. Further, the centroids are calculated, as given by Equations (7) and (8), as $\bar{p}$ and $\bar{q}$.
(7)$$\bar{p}=\frac{R_{10}}{R_{00}},$$
(8)$$\bar{q}=\frac{R_{01}}{R_{00}}.$$

The Hahn moments must be converted from 1D notation to a square matrix before they can be calculated. Discrete Hahn’s moments, also known as 2D moments, necessitate square matrix input data in a 2D structure [51]. Since these moments are orthogonal possess inverse properties, therefore, the construction of original data can be constructed using the inverse discrete Hahn moment. The aforementioned remains observed, and the positional and compositional features are somehow preserved in the measured moments [25,32,33,34,44,52] Two-dimensional input data in the form N’ is used to calculate the orthogonal Hahn moments, as seen in Equation (9).
(9)$$h_{m}^{x,y}\;\left(r,M\right)={\left(M+y-1\right)}_{m}{\left(M-1\right)}_{m}\;\sum_{s=0}^{m}-1^{s}\times \frac{{\left(-m\right)}_{s}\;{\left(-r\right)}_{s}\;{\left(2M+x+y-m-1\right)}_{s}}{{\left(M+y-1\right)}_{s}{\left(M-1\right)}_{s}}\cdot \frac{1}{s!},$$
where ‘p’ and ‘q’ (p > −1, q > −1) controlling the shape of polynomials by using the adjustable parameters. The Pochhammer symbol is defined by Equation (10), as follows:(10)$${\left(ϼ\right)}_{s}=ϼ\;\left(ϼ+1\right)\;\ldots \ldots \left(ϼ+s-1\right).$$

The equation is further simplified by the Gamma operator:(11)$${\left(ϼ\right)}_{s}=\frac{\Gamma \;\left(ϼ+s\right)}{ϼ\left(ϼ\right)}.$$

The raw values of Hahn moments are usually scaled using a square norm and weighting formula, as seen in Equation (12):(12)$$h_{m}^{x^{\sim },y}\left(r,M\right)=h_{m}^{x,y}\;\left(r,M\right)\sqrt{\frac{ϼ\left(r\right)}{s_{m}^{2}}},\;m\;=0,1,\ldots ,M-1.$$

Meanwhile, in Equation (13),
(13)$$ϼ\left(r\right)=\frac{\Gamma \;\left(p+r+q\right)\Gamma \;\left(q+r+1\right){\left(p+q+r+1\right)}_{M}\;}{{\left(p+q+2r+1\right)}_{m\;}!\left(M-r-1\right)!}.$$

Hahn moments are computed for the discrete 2D data up to the 3rd order through the following, Equation (14):(14)$$H_{\mathrm{pq}}\;\sum_{j=0}^{M-1}\cdot \sum_{i=0}^{M-1}\cdot {N^{\prime }}_{i,j}{\;h}_{p}^{x^{\sim },y}\left(j,M\right){\;h}_{q}^{x^{\sim },y}\left(i,M\right)\;,\;p,\;q\;=0,1,\ldots ,M-1.$$

The special features based on the Hahn moments are represented by H_00_, H_01_, H_10_, H_11_, H_02_, H_20_, H_12_, H_21_, H_30_, and H_03_. For each protein sequence up to the third order, we produced 10 central, 10 raw, and 10 Hahn moments and added them to the miscellaneous Super Feature Vector at random (SFV).

### 4.2. Position Relative Incident Matrix (PRIM)

The amino acids’ order and location in a protein sequence have crucial importance for the recognition of protein characteristics [47,50,53]. In any protein sequence, the relative position of an amino acid remains an essential pattern for understanding its physical properties. The Position Relative Incident Matrix (PRIM) uses a square matrix of order 20 to depict the relative location of amino acids in protein sequences, which is expressed by Equation (15):(15)$$N_{\mathrm{PRIM}}=[\begin{array}{cccccc} O_{1\to 1} & O_{1\to 2} & \cdots & O_{1\to j} & \cdots & O_{1\to 20}\\ O_{2\to 1} & O_{2\to 2} & \cdots & O_{2\to j} & \cdots & O_{2\to 20}\\ \vdots & \vdots & \cdots & \vdots & \cdots & \vdots \\ O_{i\to 1} & O_{i\to 2} & \cdots & O_{i\to j} & \cdots & O_{i\to 20}\\ \vdots & \vdots & & \vdots & & \vdots \\ O_{k\to 1} & O_{k\to 2} & \cdots & O_{k\to j} & \cdots & O_{k\to 20} \end{array}]$$

$O_{i\to j}$ represents the position of the jth amino acid for the first occurrence of the ith amino acid in the chain.

The score is of biological evolutional process accomplished by amino-acid of type ‘J’. The matrix, $N_{\mathrm{PRIM}}$ has 400 coefficients based on the relative position of amino acids occurrence.

Ten central moments, 10 raw moments, and 10 Hahn moments are calculated using the 2D $N_{\mathrm{PRIM}}$ and 30 additional special features randomly applied to the miscellaneous SFV.

### 4.3. Reverse Position Relative Incident Matrix (R-PRIM)

There are several instances of cell biology where biochemical sequences are homologous in origin. This normally occurs where a single ancestor is involved in the evolution process, and several sequences are derived from it. In such situations, using these homologous sequences has a significant impact on the classifier’s output. For the purpose of obtaining correct results, successful and efficient sequence similarity searching is carried out. In machine learning, efficiency and accuracy are urgently needed for the preciseness of feature extraction algorithms through which the most relevant features are extracted from biological data [43,47,50,53].

The methods used in R-PRIM and PRIM computations are the same, but R-PRIM is only useful for reverse protein sequence ordering. The R-PRIM computations revealed hidden trends in the data and removed ambiguities between homologous sequences. R-PRIM was created as a 20 × 20 matrix containing 400 hundred coefficients, as seen in Equation (16):(16)$$N_{R-\mathrm{PRIM}}=[\begin{array}{cccccc} B_{1\to 1} & B_{1\to 2} & \cdots & B_{1\to j} & \cdots & B_{1\to 20}\\ B_{2\to 1} & B_{2\to 2} & \cdots & B_{2\to j} & \cdots & B_{2\to 20}\\ \vdots & \vdots & \cdots & \vdots & \cdots & \vdots \\ B_{i\to 1} & B_{i\to 2} & \cdots & B_{i\to j} & \cdots & B_{i\to 20}\\ \vdots & \vdots & & \vdots & & \vdots \\ B_{k\to 1} & B_{k\to 2} & \cdots & B_{k\to j} & \cdots & B_{k\to 20} \end{array}]$$

The N_R-PRIM_ 2D matrix is used to measure 10 raw, 10 central, and 10 Hahn moments up to 3rd order, as well as more than 30 special features that are randomly applied to the SFV range.

### 4.4. Frequency Distribution Vector (FDV)

A frequency distribution vector (FDV) can be generated by using the distribution rate of each amino acid in a protein chain, as expressed in Equation (17).
(17)$$µ=\left\{a_{i},a_{2},a_{3},a_{4},\ldots \ldots \ldots \ldots \ldots a_{20}\right\}.$$

Here $a_{i}$ is the occurrence of frequency of ith (1≤ i ≤ 20) amino acid in each protein chain. Twenty more special functions have been randomly added to the SFV’s miscellany.

### 4.5. AAPIV (Accumulative Absolute Position Incidence Vector)

The AAPIV was used to retrieve relevant amino acid positional information, which retrieves and stores amino acid positional information for 20 native amino acids in a protein sequence [50,53]. This creates 20 critical features associated with each amino acid in a sequence, as expressed by Equation (18). These 20 new features were thrown into the SFV at random.
(18)$$\mathrm{AAPIV}=\left\{\mu_{1},\mu_{2},\mu_{3},\ldots ,\mu_{20}\right\},$$
where $\Phi_{i}$ is expressed by Equation (19).
(19)$$\mu_{i}=\sum_{x=1}^{n}R_{x}.$$

The $\mu_{i}$ comes from the protein sequence R_x_, which has a cumulative amino acid count of ‘n’, which can be determined using Equation (19).

### 4.6. R-AAPIV (Reverse Accumulative Absolute Position Incidence Vector)

R-AAPIV uses reverse sequence ordering to extract and store positional information of amino acids with respect to 20 native amino acids in a protein sequence, which is in reverse order relative to AAPIV [50,53].

This creates 20 critical features associated with each amino, as expressed by Equation (20).
(20)$$R-\mathrm{AAPIV}=\left\{\Phi_{i},\Phi_{2},\Phi_{3},\ldots \ldots \ldots \ldots ,\;\Phi_{20}\right\},$$
where $\Phi_{i}$ is expressed by Equation (21).
(21)$$\Phi_{i}=\sum_{x=1}^{n}\;\mathrm{Reverse}\;{\left(R\right)}_{x}.$$
where $R_{1},R_{2},R_{3},\ldots R_{n}$ are the ordinal locations at which the residue of protein sequence occurs in the reverse sequence? The values of an arbitrary element of $\Phi_{i}$ are given by Equation (21).

## 5. Machine Learning Classifiers

The Random Forest and Probabilistic Neural Network are used as a training model to predict acetylation and non-acetylation sites in this research. The following sections go into these classification algorithms in greater depth.

### 5.1. Probabilistic Neural Network

In this paper, we used a Probabilistic Neural Network (PNN) as a classifier. The Probabilistic Neural Network (PNN) is a powerful algorithm that is mostly employed for classification problems. PNN was first introduced by Specht in 1990 [54]. It is a feed-forward neural network that works on the principle of Kernel Fisher Discriminant Analysis. PNN uses the probability density function and generates better prediction results compared to other neural network algorithms [55,56].

The PNN operates on four layers, i.e., (1) input layer, (2) hidden layer, (3) pattern layer/summation layer, and (4) output layer [57]. The input layer accepts the training samples as input; further, the input data is forwarded to the hidden layer to operate the computational functions. In the hidden layer, an individual node is a computational unit that has some weighted input connections. The third layer received the probability results along with its given classes [58]. The output layer takes the decision and assigns the respective class label to the unknown sample. The schematic view of the PNN is shown in Figure 3.

### 5.2. Random Forest

In this paper, we used Random Forest (RF) as a classifier [59]. Random Forest (RF) is an ensemble learning method for classification and regression problems that is widely used [60,61,62]. RF, as its name implies, contains a large number of individual decision trees that operate as an ensemble. Each individual tree in the Random Forest splits the dataset into training and testing subsets. The training subset is used for training the model, and the testing subset is employed for testing the prediction performance of the trained model classification. During the classification, the class label is assigned to the testing sample by receiving the majority votes of all trees. The variation or bias of a single tree has little effect on the overall prediction accuracy because of the majority voting principle. RF also implements the concept of the weight model by providing a weight value that is low when a particular tree consumes a high error rate. Random Forest builds multiple decision trees and merges them together to get a more accurate and stable prediction. Random Forest has nearly the same hyper parameters as a decision tree or a bagging classifier. Random Forest adds additional randomness to the model while growing the trees [59]. The working of the Random Forest algorithm is illustrated by the following procedure.

The first step starts with the selection of random samples from a given dataset; in the second step, this algorithm will construct a decision tree for every sample, and, in the last step, voting will be performed for every predicted result [63].

We tested the proposed model performance using the different numbers of trees and found the best results when the number of trees is 500. The RF offers several advantages, including its optimal accuracy, it works efficiently with large datasets, and its detection of outliers. Many researchers used RF for solving several biological problems and achieve better performance. The working mechanism of the RF is depicted in Figure 4.

## 6. Performance Evaluation Parameters and Testing Methods

### 6.1. Performance Evaluation Parameters

The performance of the proposed model can be measured by effective evaluation metrics. We use the subsequent four metrics to measure the forecast quality: (1) Overall Accuracy (ACC), (2) Sensitivity (Sn), (3) Specificity (Sp), and (4) Mathews Correlation Coefficient (MCC). We computed these parameters using a binary confusion matrix. These metrics remain the most common metrics used to measure efficiency of the proposed model. We computed these parameters using the following equations.
(22)$$\{\begin{array}{c} \mathrm{ACC}=\frac{\mathrm{TP}+\mathrm{TN}}{\mathrm{TP}+\mathrm{TN}+\mathrm{FP}+\mathrm{FN}}\\ S_{n}=\frac{\mathrm{TP}}{\mathrm{TP}+\mathrm{FN}}\\ S_{p}=\frac{\mathrm{TN}}{\mathrm{TN}+\mathrm{FP}}\\ \mathrm{MCC}=\frac{\left(\mathrm{TP}\times \mathrm{TN})-(\mathrm{FP}\times \mathrm{FN}\right)}{\sqrt{\left(\mathrm{TP}+\mathrm{FP})(\mathrm{TP}+\mathrm{FN})(\mathrm{TN}+\mathrm{FP})(\mathrm{TN}+\mathrm{FN}\right)}},\\ \mathrm{Recall}=\frac{\mathrm{TP}}{\mathrm{TP}+\mathrm{FN}}\\ \mathrm{Precision}=\frac{\mathrm{TP}}{\mathrm{TP}+\mathrm{FP}}\\ F.\mathrm{measure}=2\times \frac{\mathrm{Recall}\times \mathrm{Precision}}{\mathrm{Recall}+\mathrm{Precision}} \end{array}$$
where TP is True Positive, FN is False Negative, TN is True Negative, and FP is False Positive, respectively.

As per the given confusion matrix as shown in the Table 1, we subsequently calculate the following:

TP: Production prognosis, such as True Positive (TP), where we found that acetylation subject stays properly categorized, as well as classified, then the subjects have acetylation proteins.

TN: Production forecast, such as True Negative (TN), where we found that an non-acetylation protein remains properly classified, and then the subject remains non-acetylation protein.

FP: Production prognosis, such as false positive (FP), where we found that non- acetylation protein remains inaccurately classified as containing acetylation proteins known as “type 1 error”.

FN: Forecast of production, such as false negative (FN), where we found that acetylation proteins remain inaccurately classified and that the subject has non- acetylation proteins, this is the “type 2 error”.

ROC and AUC: The optimistic receiver curves evaluate the predictability of the machine learning classifiers at various threshold settings. The ROC exam remains a graphical demonstration that relates the “true positive rate” to the “false positive rate” in the grouping results of the machine learning algorithm. AUC describes a classifier of the ROC. The higher value of the AUC more than 0.5, suggest discrimination, whereas the value of 0.5 doesn’t suggest any discrimination in true positive and true negative of classifier, more the AUC value more the efficiency in the performance of a classifier.

### 6.2. Testing Methods

Several cross-validation techniques have been used to examine the statistical forecaster’s results in the literature. The jackknife test, independent dataset test, and k-fold cross-validation test are three experiments that are commonly used by various researchers.

When testing a forecaster designed for its efficiency, we use the following cross-validation methods in this paper to estimate the expected accuracy of the forecaster, self-consistency, independent, K-fold cross-validation, and jackknife testing for the assessment of the proposed model.

The following sub-sections contain the details.

### 6.3. K-Fold (10-Fold) Cross-Validation Test

The K-fold cross-validation (KFCV) test is a technique to estimate predictive models by partitioning the original dataset A into disjoint k-folds {A_1_, A_2_, A_3_, A_i_…, A_k_}, where it uses A-A_i_ folds for training, and the remaining ith fold A_i_ for testing, where i = 1, 2, 3, …k as shown in the Figure 5. The method iterates the process for each i, and calculate the performance that is the accuracy, sensitivity, precession, recall, F-Measure, and MCC. Further, for the overall result, the average is taken of all the iterations performed for each fold. This technique has many benefits, such as the fact that it validates the model based on multiple datasets to reduce the bias and reach to a stable evaluation that how the model performs. This technique is much more powerful compared to other cross-validation techniques. In literature, the K-fold method is quite popular for k = 10 and 5. In this research, a 10-fold cross-validation is used: the overall result obtained is 100% with random forest classifier as presented in Table 2, and through ROC curves shown in Figure 6. 

The results obtained from three dataset using 10-folds Cross validation with the Random forest classifier, and achieved the best result with ACC 100%, MCC, Sn, Precision, and F-measure all are 1.

We also used a 10-fold cross-validation test to evaluate the PNN-based model on the three datasets, and obtained the ACC 66.83, MCC 0.36, Sn 0.72, Precision 0.65, and F-measure 0.72 as presented in the Table 3 and by the ROC curve in the Figure 7.

### 6.4. Jackknife Test

Several cross-validation tests are extensively useful to estimate the performance of the statistical predictors. Amongst these, the jackknife test is considered to be the supreme in being consistent and reliable. Consequently, the jackknife test is comprehensively applied by researchers to estimate the performance of the predictor model. In this test, if the dataset has N records in the dataset, then it trains the model for N − 1 records and tests the model for the remaining one record, which is why it is also called leave-one-out cross-validation. Further, this process is repeated N-times, and the label of each record is predicted. Finally, we accumulate all the results to make the overall prediction based on accuracy, sensitivity, precession, recall, F-measure, and MCC.

In present research work, the jackknife test is used to measure the performance of models by using the classifiers Random Forest and PNN, and we achieved the result of 100% through RF but got 66.87 through PNN, as presented in Table 4 and Table 5, and by the ROC curves in Figure 8 and Figure 9.

In this evaluation process, three datasets were used which give the overall accuracy, sensitivity, specificity, precession, MCC, Recall, and F-Measure, given in detail in Table 4.

### 6.5. Self-Consistency Test

Self-consistency test is a technique referred to as the ultimate test for the validation of efficiency and efficacy of the prediction model, this method uses the same data for both training and testing A representation of these proposed parameters, by conducting the self-consistency testing, the results for the acetylation protein prediction based on the Random Forest classifier as presented in the Table 6 and by the ROC curve is shown in the Figure 10.

Similarly, the evaluation results based on the PNN for the three datasets, S1, S2, and S3, based on RF and PNN as presented in the Table 7 and by the ROC curve as shown in the Figure 11.

### 6.6. Independent Test

An independent test is a cross-validation method that objectively finds out the predictor’s performance metrics of the planned model, which obtains values from a confusion matrix to evaluate the performance of the model. In this method, the dataset is divided into two parts, training, and the testing part. In the proposed work the data is split in two parts that is 70% of the data for the training and the remaining 30% for the testing as shown in Figure 12. The method used to train and test the models based on the classifiers, the Random Forest and the PNN, and we obtained the result of 98% through RF, and 50.8 through PNN. The area under the curve (AUC), obtained by Random Forest and PNN, is 98% and 50.8%, respectively. The remaining detailed results, based on the two classifiers, is presented in Table 8 and Table 9 and by the ROC curve in Figure 13 and Figure 14, respectively.

## 7. Results and Discussions

The most critical thing is to compare the proposed novel model to other state-of-the-art models in order to assess its prediction accuracy. When compared to well-known current classifiers, the RF and the PNN. In this work, the model with RF achieved significantly higher accuracy and efficiency in predicting acetylation from non-acetylation, as seen in Table 10 and by Figure 15 and Figure 16.

### Comparison to Existing Models

To show the efficiency of our proposed model, iAcety–SmRF, which achieved the highest accuracy of 100% with Sensitivity, Precision, and MCC, is 1, as shown in Section 4.2 (B) (i), using a 10-fold cross-validation. The proposed model was compared to several models, including the latest iAcet-PseFDA by Reference [1]; they used the same dataset and developed a method for predicting acetylation proteins by extracting features from sequence conservation information using a gray frame model and an ANN score based on information from the annotation of the functional domains and subcellular localization. For model validation, they used a 5-fold cross-validation for all three datasets and achieved an average accuracy of 77.10%. The All-Mean JK model by Nakai and Horton [36] achieved an accuracy of 74.64%. Hunter constructed a predictive model InterPro, with accuracy of 68.25% reference [64]. Table 11 lists all comparative analyses of the proposed study, based on our two models iAcety-SmRF and iAcety-SmPNN, in which iAcety-SmRF achieved the superior accuracy as compared to all existing models.

## 8. Conclusions

A computational model for predicting Acetylation sites from non-Acetylation sites is developed in this paper. The model contained Statistical Moments used for extraction of features into the equivalent numerical form of the original biological data. Further, Random Forest and Probabilistic Neural Network were applied for its classification to predict acetylation from non-acetylation. Furthermore, independent testing, 10-fold cross-validation, self-consistency test, and jackknife testing are used to evaluate accuracy, with results of 97%, 100%, and 100%, respectively, based on the Random Forest. Further, the model was compared with the already existing relevant models available in the literature, which revealed the remarkable performance of our work. Finally, the final step is to develop an influential website/GitHub resource for public use for the benefit of future research and development which can be accessed by the following link: https://github.com/shaistarahmanmcs/My-Website-identifying-Proteins-Acetylation-.git (accessed on 8 December 2021).

## Figures and Tables

**Figure 1 membranes-12-00265-f001:**
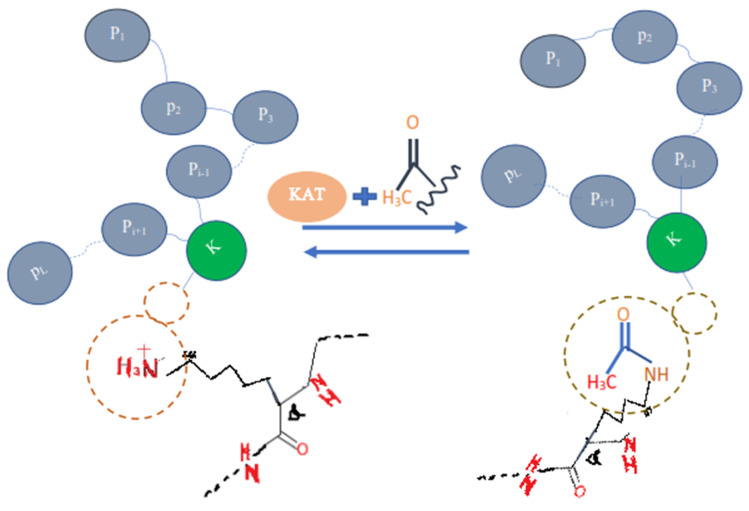
Acetylation protein.

**Figure 2 membranes-12-00265-f002:**
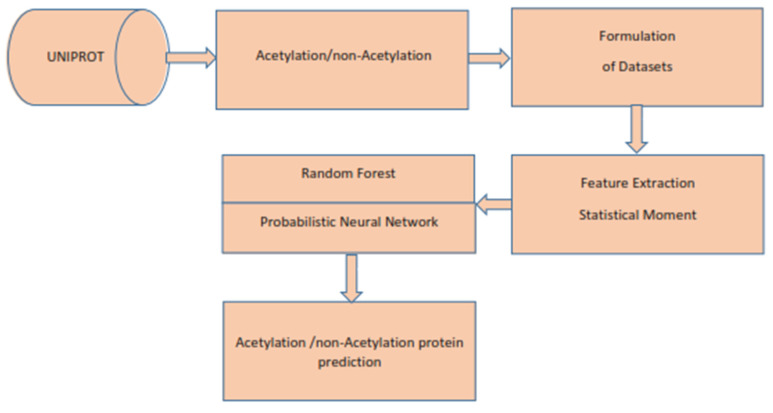
Flowchart of the proposed predictor.

**Figure 3 membranes-12-00265-f003:**
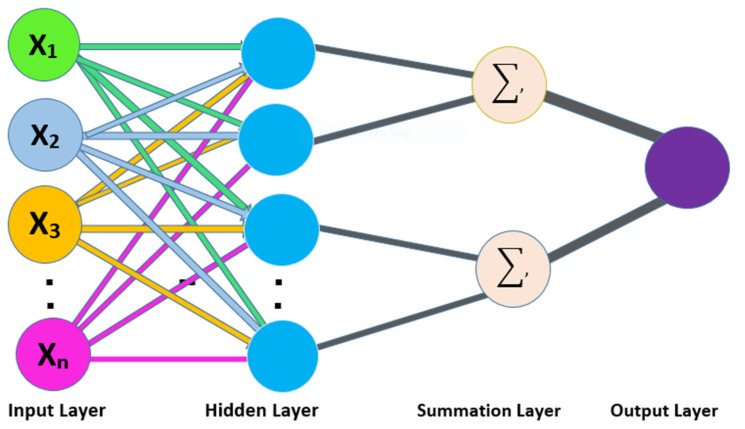
Schematic view of PNN.

**Figure 4 membranes-12-00265-f004:**
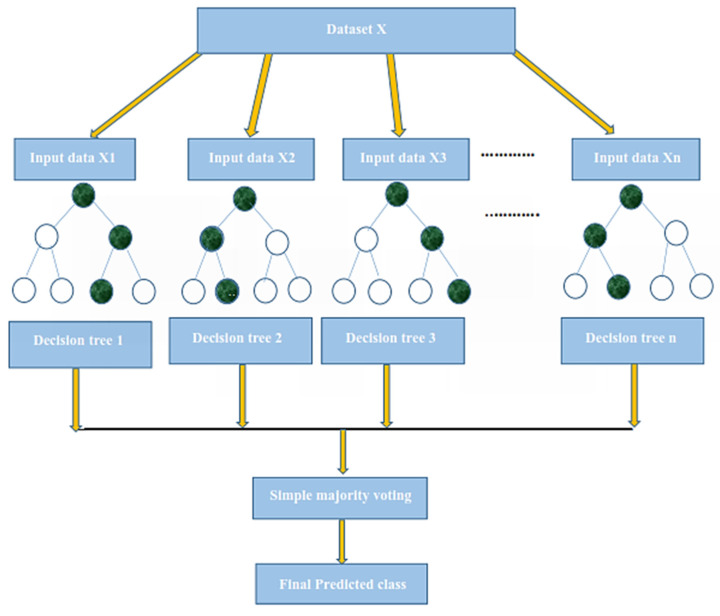
The working mechanism of the Random Forest classifier.

**Figure 5 membranes-12-00265-f005:**
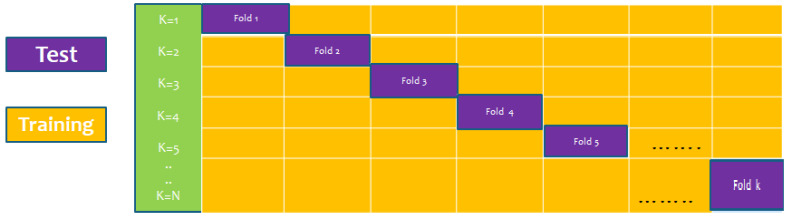
K-fold cross-validation (KFCV).

**Figure 6 membranes-12-00265-f006:**
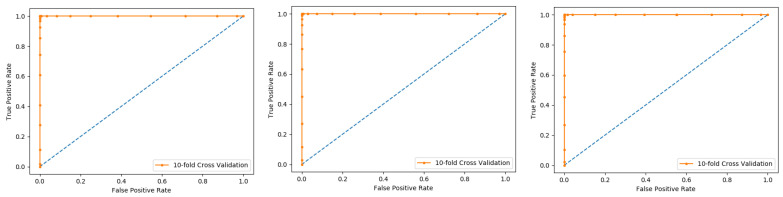
10-fold Random Forest ROC curve.

**Figure 7 membranes-12-00265-f007:**
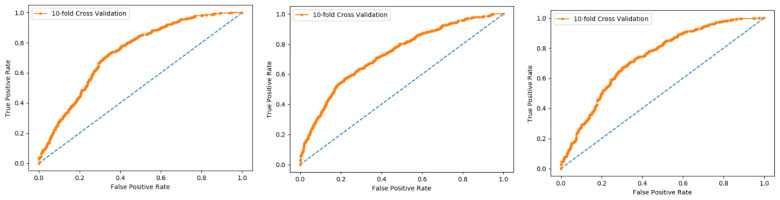
Ten-fold Probabilistic Neural Network ROC curve.

**Figure 8 membranes-12-00265-f008:**
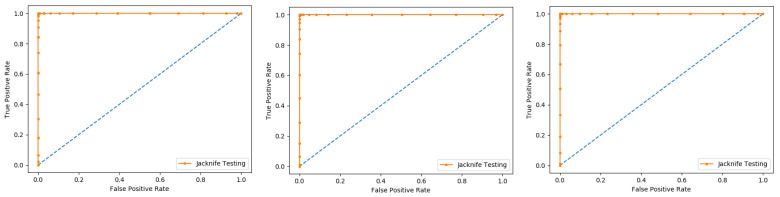
Jackknife Random Forest ROC curve.

**Figure 9 membranes-12-00265-f009:**
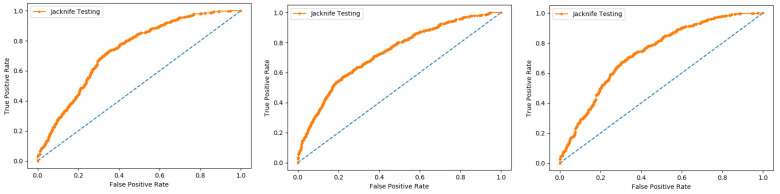
Jackknife PNN ROC curve.

**Figure 10 membranes-12-00265-f010:**
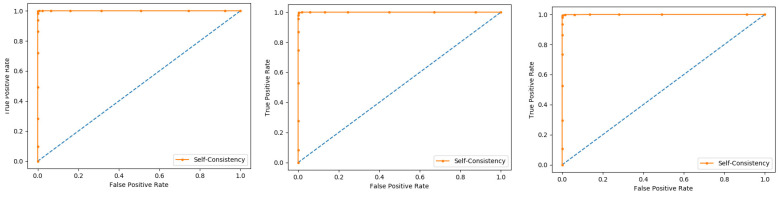
Self- consistency Random Forest ROC curve.

**Figure 11 membranes-12-00265-f011:**
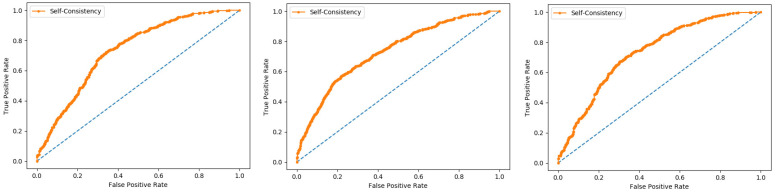
Self- consistency Probabilistic Neural Network ROC curve.

**Figure 12 membranes-12-00265-f012:**
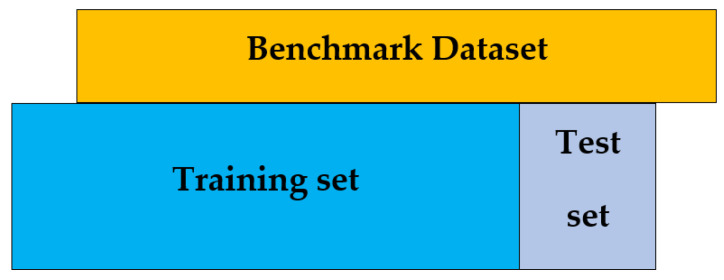
Independent test.

**Figure 13 membranes-12-00265-f013:**
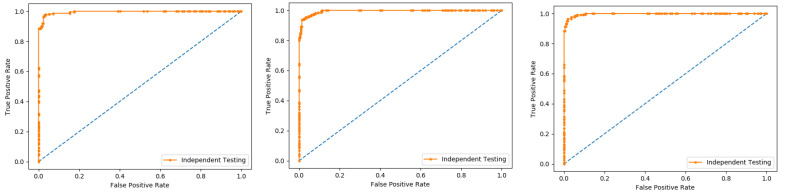
Independent Random Forest ROC curve.

**Figure 14 membranes-12-00265-f014:**
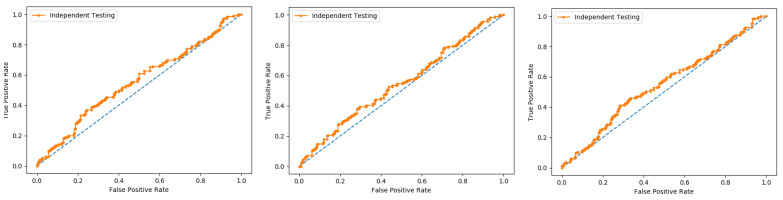
Independent Probabilistic Neural Network ROC curve.

**Figure 15 membranes-12-00265-f015:**
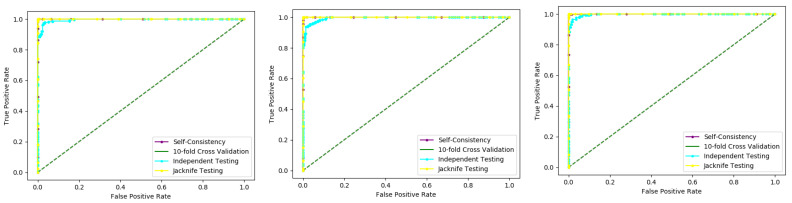
ROC curves through Random Forest.

**Figure 16 membranes-12-00265-f016:**
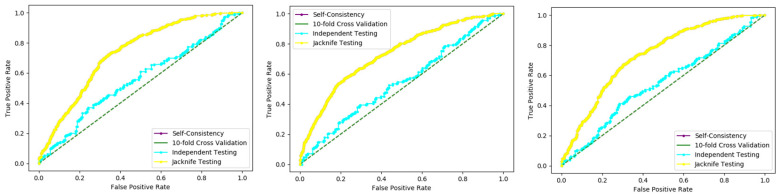
ROC curves through Probabilistic Neural Network.

**Table 1 membranes-12-00265-t001:** Confusion matrix.

Status Person	Predicted Patient (1)	Predicted Healthy Person (0)
Actual patient (1)	TP	FN
Actual healthy person (0)	FP	TN

**Table 2 membranes-12-00265-t002:** Result of 10-fold cross-validation based on the Random Forest classifier.

10-Fold Cross-Validation Random Forest																					
	Dataset 1							Dataset 2							Dataset 3						
K-Folds	ACC	Sn	Sp	Prec	MCC	Recall	F.m	ACC	Sn	Sp	Prec	MCC	Recall	F.m	ACC	Sn	Sp	Prec	MCC	Recall	F.m
1	100	1	1	1	1	1	1	100	1	1	1	1	1	1	100	1	1	1	1	1	1
2	100	1	1	1	1	1	1	100	1	1	1	1	1	1	100	1	1	1	1	1	1
3	100	1	1	1	1	1	1	100	1	1	1	1	1	1	100	1	1	1	1	1	1
4	100	1	1	1	1	1	1	100	1	1	1	1	1	1	100	1	1	1	1	1	1
5	100	1	1	1	1	1	1	100	1	1	1	1	1	1	100	1	1	1	1	1	1
6	100	1	1	1	1	1	1	100	1	1	1	1	1	1	100	1	1	1	1	1	1
7	100	1	1	1	1	1	1	100	1	1	1	1	1	1	100	1	1	1	1	1	1
8	100	1	1	1	1	1	1	100	1	1	1	1	1	1	100	1	1	1	1	1	1
9	100	1	1	1	1	1	1	100	1	1	1	1	1	1	100	1	1	1	1	1	1
10	100	1	1	1	1	1	1	100	1	1	1	1	1	1	100	1	1	1	1	1	1
result	100	1	1	1	1	1	1	100	1	1	1	1	1	1	100	1	1	1	1	1	1

**Table 3 membranes-12-00265-t003:** 10-fold cross-validation Result for Probabilistic neural network.

	10-Fold Cross-Validation Probabilistic Neural Network																				
	Dataset 1							Dataset 2							Dataset 3						
K-Folds	ACC	Sn	Sp	Prec	MCC	Recall	F.m	ACC	Sn	Sp	Prec	MCC	Recall	F.m	ACC	Sn	Sp	Prec	MCC	Recall	F.m
Final Score	66.83	0.72	0.60	0.65	0.36	0.95	0.72	60	0.26	0.93	0.81	0.21	0.26	0.40	57.17	0.92	0.22	0.54	0.36	0.92	0.72

**Table 4 membranes-12-00265-t004:** Jackknife test score based on Random Forest.

Predicton	Jackknife Random Forest																				
	Dataset 1							Dataset 2								Dataset 3					
Fold	ACC	Sn	Sp	Pre	MCC	Recall	F.m	ACC	Sn	Sp	Prec	MCC	Recall	F.m	ACC	Sn	Sp	Prec	MCC	Recall	F.m
Result	100	0.55	0.5	0.55	0.05	0.003	0.01	99.86	0.55	0.5	0.55	0.05	0.54931	0.55	99.86	0.55	0.5	0.55	0.05	0.54931	0.5

**Table 5 membranes-12-00265-t005:** Jackknife score based on the Probabilistic Neural Network.

Prediction	Jackknife Probabilistic Neural Network																				
	Dataset 1							Dataset 2							Dataset 3						
Jackknife	ACC	Sn	Sp	Prec	MCC	Recall	F.m	ACC	Sn	Sp	Prec	MCC	Recall	F.m	ACC	Sn	Sp	Prec	MCC	Recall	F.m
Final Score	66.87	0.55	0.5	0.5	0.6	0.41	0.42	59.77	o.5	0.5	0.5	0.6	0.13	0.20	57.41	0.55	0.4	0.5	0.6	0.50	0.48

**Table 6 membranes-12-00265-t006:** Results of self-consistency based on Random Forest.

Self-Consistency Random Forest																				
Dataset 1							Dataset 2							Dataset 3						
ACC	Sn	Sp	Prec	MCC	Recall	F.m	ACC	Sn	Sp	Prec	MCC	Recall	F.m	ACC	Sn	Sp	Prec	MCC	Recall	F.m
100	1	0.99	1	0.99	0.997	1	100	1	1	1	0.99	0.997	1	100	1	1	1	1	1	1

**Table 7 membranes-12-00265-t007:** Self-consistency test result for probabilistic neural network.

Self-Consistency Probabilistic Neural Network																				
Dataset 1							Dataset 2							Dataset 3						
ACC	Sn	Sp	Prec	MCC	Recall	F.m	AC	Sn	Sp	Prec	MCC	Recall	F.m	ACC	Sn	Sp	Prec	MCC	Recall	F.m
66.83	0.72	0.60	0.64	0.36	0.72	0.68	60	0.26	0.93	0.80	0.20	0.26	0.39	57.17	0.92	0.22	0.54	0.36	0.92	1.84

**Table 8 membranes-12-00265-t008:** Results of independent test based on the Random Forest.

Independent Test Results Random Forest																				
Dataset 1							Dataset 2							Dataset 3						
Training Dataset							Training Dataset							Training Dataset						
ACC	Sn	Sp	Prec	MCC	Recall	F.m	ACC	Sn	Sp	Prec	MCC	Recall	F.m	ACC	Sn	Sp	Prec	MCC	Recall	F.m
97	1	1	1	1	0.969	1	96	1	1	1	1	0.97	1	96	1	1	1	1	0.95	1
Testing Dataset							Testing Dataset							Testing Dataset						
ACC	Sn	Sp	Prec	MCC	Recall	F.m	ACC	Sn	Sp	Prec	MCC	Recall	F.m	ACC	Sn	Sp	Prec	MCC	Recall	F.m
98	1	1	1	1	0.96	1	95	1	1	1	1	0.93	1	97	1	1	1	1	0.96	1

**Table 9 membranes-12-00265-t009:** Result of independent test based on Probabilistic Neural Network.

Independent Test Result Probabilistic Neural Network																			
Dataset 1						Dataset 2							Dataset 3						
**Independent Training Dataset Confusion Matrix**						**Independent Training Dataset Confusion Matrix**							**Independent Training Dataset Confusion Matrix**						
ACC	Sn	Prec	MCC	Recall	F.m	ACC	Sn	Sp	Prec	MCC	Recall	F.m	ACC	Sn	Sp	Prec	MCC	Recall	F.m
50.8	0.1	1	0.6	0.115	0.20	52.6	0.7	0.3	0.53	0.05	0.75	0.62	51.33	1	0.1	1	0.5	0.97	0.98
**Independent Testing Dataset Confusion Matrix**						**Independent Testing Dataset Confusion Matrix**							**Independent Testing Dataset Confusion Matrix**						
ACC	Sn	Prec	MCC	Recall	F.m	ACC	Sn	Sp	Prec	MCC	Recall	F.m	ACC	Sn	Sp	Prec	MCC	Recall	F.m
50.8	1	1	0.6	0.92	0.96	54.0	0.2	0.9	0.54	0.06	0.19	2.86	51.03	1	0.9	1	0.6	0.07	0.13

**Table 10 membranes-12-00265-t010:** Performance of proposed model based on RF and PNN.

Prediction	Comparative Analysis of RF and PNN																				
	Dataset 1							Dataset 2							Dataset 3						
Classifier	ACC	Sn	Sp	Prec	MCC	Recall	F.m	ACC	Sn	Sp	Prec	MCC	Recall	F.m	ACC	Sn	Sp	Prec	MCC	Recall	F.m
RF	100	1	1	1	1	1	1	1	1	1	1	1	1	1	1	1	1	1	1	1	1
PNN	66.83	0.72	0.6	0.65	0.36	0.95	0.72	60	0.26	0.93	0.81	0.21	0.26	0.40	57.17	0.92	0.22	0.54	0.36	0.92	0.72

**Table 11 membranes-12-00265-t011:** Comparative analysis of the proposed acetylation model with the existing models.

Prediction Models	ACC%	MCC%	Sn%	Sp%	Prec%	F.m%
All-Mean JK	74.64%	0.4980	81.38%	67.91%	71.78%	76.24%
iAcet-PseFDA	77.55%	0.5883	96.41%	71.26%	52.79%	68.23%
InterPro	68.25%	0.3658	71.40%	65.10%	67.17%	69.22%
iAcety–SmRF	100	1.0	1.0	1.0	1.0	1.0
iAcety–SmPNN	66.83	0.36	0.72	0. 60	0.65	0.72

## Data Availability

The dataset and classification code is available on the GitHub: https://github.com/shaistarahmanmcs/My-Website-identifying-Proteins-Acetylation-.git (accessed on 8 December 2021).

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
