# Peer review of "iAcety–SmRF: Identification of Acetylation Protein by Using Statistical Moments and Random Forest"

_membranes, 2022, doi:10.3390/membranes12030265_

Round 1

Reviewer 1 Report

The authors investigated an efficient and accurate computational method to predict acetylation using machine learning approaches. The proposed model achieved an accuracy of 100 percent using the 10-fold cross-validation test based on the Random Forest classifier along with a feature extraction approach using statistical moments.
1. The related work section needs to be improved to add more related work as well as to make the knowledge gap more clear.
2. Authors need to explain Table 11, like how variations of datasets, resource requirement etc. can affect their comparison.
3. In addition to figure titles, figure legends should be available that explain the meaning of the figure. Some of the figures are not clear. Similarly, equations may also be well formatted.
4. Materials and Methods sections needs improvement. Authors did not describe the datasets. How they constructed the data, including preprocessing etc.

Author Response

Dear Editor, Thanks to reviewer for their comments and suggestions, we tried to update the research paper acoordingly. The following are the reviewer’s comments and our response in the bold text.

Response to Review-1

English language and style

( ) Extensive editing of English language and style required
(x) Moderate English changes required

Reply: updated
( ) English language and style are fine/minor spell check required
( ) I don't feel qualified to judge about the English language and style

Yes

Can be improved

Must be improved

Not applicable

Does the introduction provide sufficient background and include all relevant references?

(x)

( )

( )

( )

Is the research design appropriate?

(x)

( )

( )

( )

Are the methods adequately described?

(x)

( )

( )

( )

Are the results clearly presented?

Reply: it is updated

( )

(x)

( )

( )

Are the conclusions supported by the results?

(x)

( )

( )

( )

Comments and Suggestions for Authors

The authors investigated an efficient and accurate computational method to predict acetylation using machine learning approaches. The proposed model achieved an accuracy of 100 percent using the 10-fold cross-validation test based on the Random Forest classifier along with a feature extraction approach using statistical moments.
1. The related work section needs to be improved to add more related work as well as to make the knowledge gap moreclear.

Reply. 1. The last research published for the prediction of acetylation protein was conducted by Qiu et al.

Qiu, W. R., Xu, A., Xu, Z. C., Zhang, C. H., & Xiao, X. (2019). Identifying acetylation protein by fusing its PseAAC and functional domain annotation. Frontiers in bioengineering and biotechnology, 7, 311. Which is cited

The maximum accuracy achieved by them is 77.10. Further, all the previous predictors of acetylation protein are cited. To make knowledge clearer we included the paragraph on page 6 with red text.

  1. Authors need to explain Table 11, like how variations of datasets, resource requirement etc. can affect their comparison.

Reply: the complete details as red text is given on section V(I) on page 28

  1. In addition to figure titles, figure legends should be available that explain the meaning of the figure. Some of the figures are not clear. Similarly, equations may also be well formatted.

Reply: All figures captions are checked and updated, are updated.

4. Materials and Methods sections needs improvement. Authors did not describe the datasets. How they constructed the data, including preprocessing etc.

Reply: Materials and Methods sections is checked and updated. The dataset is described explained the preprocessing as highlighted with red color text

Reviewer 2 Report

The manuscript presents a new model to predict protein acetylation through machine learning approaches.  In general, the work is well done showing a new strategy using statistical moments and random forest. It can be of interest to those working in machine learning and specifically to those involved in protein acetylation. Just a couple of things before its acceptance for publication.

1.- A general editing of the manuscript is suggested because it is not easy to read in several parts and there are some typos.

2.- According to the comparisons with existing models, the model presented here is better, however, a deeper discussion about this is necessary, not only show the results obtained with each model.

Author Response

Dear Editor, Thanks to reviewer for their comments and suggestions, we tried to update the research paper accordingly. The following are the reviewer’s comments and our response in the bold text.

Response to Review-2

Open Review

English language and style

( ) Extensive editing of English language and style required
(x) Moderate English changes required

Reply: It is updated
( ) English language and style are fine/minor spell check required
( ) I don't feel qualified to judge about the English language and style

Yes

Can be improved

Must be improved

Not applicable

Does the introduction provide sufficient background and include all relevant references?

( )

(x)

( )

( )

Is the research design appropriate?

( )

(x)

( )

( )

Are the methods adequately described?

( )

(x)

( )

( )

Are the results clearly presented?

( )

(x)

( )

( )

Are the conclusions supported by the results?

( )

(x)

( )

( )

Comments and Suggestions for Authors

The manuscript presents a new model to predict protein acetylation through machine learning approaches.  In general, the work is well done showing a new strategy using statistical moments and random forest. It can be of interest to those working in machine learning and specifically to those involved in protein acetylation. Just a couple of things before its acceptance for publication.

1.- A general editing of the manuscript is suggested because it is not easy to read in several parts and there are some typos.

Reply: We tried to update the manuscript for its flow of English, however, if accepted and needed more correction then we will go for the paid editing services

2.- According to the comparisons with existing models, the model presented here is better, however, a deeper discussion about this is necessary, not only show the results obtained with each model.

Reply: the complete details as red text is given on section V(I) on page 28

Round 2

Reviewer 2 Report

The manuscript was corrected according to the suggestions, therefore, I recommend its publication.

Author Response

Thanks for the positive response, I am attaching the same reply as I uploaded for the reviewer 1